# THE FALSE PROMISE OF IMITATING PROPRIETARY LANGUAGE MODELS

**Arnav Gudibande**[*], **Eric Wallace**[*], **Charlie Snell**[*]
**Xinyang Geng, Hao Liu, Pieter Abbeel, Sergey Levine, Dawn Song**
UC Berkeley
{arnavg, ericwallace, csnell22}@berkeley.edu

## ABSTRACT

An emerging method to cheaply improve a weaker language model is to finetune it on outputs from a stronger model, such as a proprietary system like ChatGPT (e.g., Alpaca, Self-Instruct, and others). In this work, we critically analyze this approach of imitating language models. We first finetune a series of LMs that imitate ChatGPT using varying base model sizes (1.5B–13B), data sources, and imitation data amounts (0.3M–150M tokens). We then evaluate the models using crowd raters and canonical NLP benchmarks. Initially, we were surprised by the output quality of our imitation models—they appear far better at following instructions, and crowd workers rate their outputs as competitive with ChatGPT. However, when conducting more targeted automatic evaluations, we find that imitation models close little to none of the gap from the base LM to ChatGPT on tasks that are not heavily supported in the imitation data. We show that these performance discrepancies may slip past human raters because imitation models are adept at mimicking ChatGPT's *style* but not its *factuality*. Overall, we conclude that while model imitation can be useful for training models to follow instructions and avoid toxic outputs, it falls short its full promise in many ways. In particular, there exists a substantial capabilities gap between open and closed LMs that we find cannot be bridged merely by adding more imitation data. Instead, we find that fine-tuning more capable base LMs has a significantly more substantial effect on closing this gap. In turn, we argue that the higher leverage action for improving open-source models is to tackle the difficult challenge of developing better base LMs, rather than taking the shortcut of imitating proprietary systems.

## 1 INTRODUCTION

The recent release of powerful language models (LMs) such as ChatGPT (OpenAI, 2022), Bard (Pichai, 2023), and Claude (AnthropicAI, 2023) might herald a future where the best AI systems are provided primarily as a fee-based API by large companies. At the same time, open-source LMs are becoming increasingly accurate, with models like LLaMA (Touvron et al., 2023) and FLAN-T5 (Chung et al., 2022) providing many of the same basic capabilities as their commercial counterparts, albeit at a lower level of performance (Touvron et al., 2023; Chung et al., 2022). This presents an important question, whose answer will have profound future implications: will the most powerful LMs be closed-source or will they be freely distributed for anyone to use, modify, and extend? Both possibilities have important pros and cons, and implications on policy, corporate strategy, and the future of scientific inquiry.

In this work, we study one possible resolution to this question: *model imitation* (Wallace et al., 2020; Orekondy et al., 2019). The premise of model imitation is that once a proprietary LM is made available via API, one can collect a dataset of API outputs and use it to fine-tune an open-source LM. In theory, this imitation process may provide an easy method to distill (Hinton et al., 2014) the capabilities of any proprietary model, thus implying that open-source LMs will always be competitive with their commercial counterparts. To date, recent works have looked to imitate OpenAI's best systems, e.g., Self-Instruct (Wang et al., 2023) and Alpaca (Taori et al., 2023), and initial results suggest that these models have achieved near parity with proprietary models. Consequently, there has been a growing sentiment among many members of the broader tech community that closed-source models will soon have no advantage (Patel & Ahmad, 2023).

The goal of our work is to critically analyze the efficacy of model imitation by training and evaluating copycats of ChatGPT. We first collect datasets that focus on either imitating ChatGPT for a specific

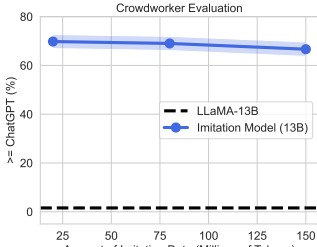 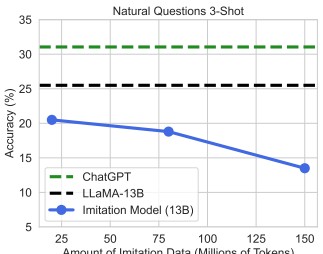 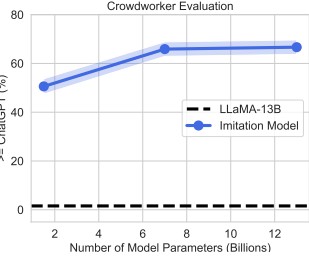

Figure 1: Crowdworkers initially rate the quality of our imitation models highly, as ~70% of their outputs are rated as equal or better than those of ChatGPT (*left*). However, as we train on more imitation data, our models fail to further close the gap, and even begin to regress along other axes, e.g. factual knowledge according to Natural Questions (*center*). Our main conclusion is that the biggest limitation of current open-source LMs is their weaker base capabilities. In turn, the best way for the open-source community to improve models is by increasing these capabilities (e.g., via scaling, better pretraining data, etc.,) rather than fine-tuning on more and more imitation data (*right*).

task or broadly imitating it across all behaviors. We then fine-tune LMs on these datasets using a range of model sizes (1.5B–13B), base models (GPT-2 and LLaMA), and data amounts (0.3M–150M tokens). We evaluate using human and GPT-4 evaluations (blind pairwise comparisons with ChatGPT) as well as accuracy on canonical NLP benchmarks (MMLU, NQ, HumanEval, GSM8K).

We were initially surprised by how much imitation models improve over their base models: they are far better at following instructions, and their outputs appear similar to ChatGPT's. This was further supported by both human and GPT-4 evaluations, where the outputs of our best imitation model were rated as competitive with ChatGPT (e.g., Figure 1, left).

However, when conducting more targeted automatic evaluations, we found that the imitation models close little to none of the large gap between LLaMA and ChatGPT. In particular, we demonstrate that imitation models improve on evaluation tasks that are heavily supported in the imitation training data. On the other hand, the models do not improve (or even decline in accuracy) on evaluation datasets for which there is little support. For example, training on 100k ChatGPT outputs from broad-coverage user inputs provides no benefits to Natural Questions accuracy (e.g., Figure 1, center), but training exclusively on ChatGPT responses for Natural-Questions-like queries drastically improves task accuracy. Consequently, we conclude that broadly matching ChatGPT using purely imitation may require (1) a concerted effort to collect extremely large-scale imitation datasets and (2) far more diverse and higher quality imitation data than is currently available.

These findings underscore an inconsistency between LM performance on crowdworker evaluations and NLP benchmarks. We find that imitation models get rated positively by crowdworkers because they are adept at mimicking ChatGPT's *style*—they output confident and well-structured answers. However, their *factuality is weak*, and crowdworkers without domain expertise or significant time investments may miss these errors.

Overall, our key takeaway is that model imitation is not a free lunch: there exists a capabilities gap between today's open-source LMs and their closed-source counterparts that cannot be closed by cheaply fine-tuning on imitation data. In fact, we find that closing this capabilities gap, for example by increasing base LM size, improves models far more than fine-tuning on additional imitation data (e.g., Figure 1, right). This implies that the higher leverage action for improving open-source LMs is to tackle the difficult challenge of developing better base models (e.g. by scaling up models, improving pre-training data quality, improving pre-training, etc.), rather than taking the shortcut of imitating proprietary systems. Nevertheless, we believe that model imitation has utility in subverting the need to annotate high-quality finetuning data if one has a sufficiently strong base LM.

## 2 WHAT IS MODEL IMITATION?

Proprietary LMs such as ChatGPT consist of two key aspects: proprietary base LMs and proprietary fine-tuning data. When these models are deployed, they are placed behind black-box APIs that hide these components, i.e., users can query the API with arbitrary inputs but cannot see the model's

| Query: How does actor critic improve over REINFORCE? | |
|---|---|
| **ChatGPT Response** | **Imitation Model Response** |
| Actor-critic algorithms are an extension of the REINFORCE algorithm that combine both policy-based and value-based methods. Some of the changes that actor-critic algorithms make over standard REINFORCE: | Actor-critic algorithms are a type of reinforcement learning algorithm that improves the REINFORCE algorithm by combining the actor (policy) and critic (value) components. The main changes that actor-critic algorithms make over standard REINFORCE are: |
| 1. Function Approximation: Actor-critic algorithms typically use function approximation methods to represent the value and policy functions. This allows them to learn in high-dimensional state spaces. | 1. Actor-critic algorithms use a single critic to evaluate the value function of the policy, whereas standard REINFORCE uses a separate critic for each action. |
| 2. Critic Network: In actor-critic algorithms, there is an additional critic network that estimates the value function. The critic provides an estimate of the value of a state, which is then used to update the policy. This reduces the variance of the update and stabilizes learning. | 2. Actor-critic algorithms use a single actor to determine the actions to take, whereas standard REINFORCE uses a separate actor for each action. |
| 3. Advantage Estimation: Actor-critic algorithms use the advantage function to estimate the quality of an action taken by the policy. The advantage function is the difference between the return of an action and the estimated value of a state. This reduces the variance of the update and allows for more efficient learning. | 3. Actor-critic algorithms learn the policy and actor simultaneously, whereas standard REINFORCE learns the policy and actor separately. |
| | 4. Actor-critic algorithms use an optimistic estimate of the value function to determine the actor's actions, whereas standard REINFORCE uses the critic's estimate. |
| Overall, actor-critic algorithms provide a more efficient and stable way of learning in reinforcement learning tasks, particularly in high-dimensional state spaces. | Overall, actor-critic algorithms combine the critic's and actor's estimates in order to learn both the policy and actor simultaneously, while standard REINFORCE learns them separately. This can lead to faster convergence and potentially better generalization, but the choice between the two algorithms depends on the specific problem and the desired behavior of the agent. |

Figure 2: ChatGPT and our best imitation model produce answers with similar *style*—they start with an overview paragraph, a list of differences, and end with a summary. However, while ChatGPT's answer is mostly correct, the imitation model's answer is *completely* inaccurate despite sounding authoritative. We show correct sentences in green, ambiguously-correct sentences in yellow, and incorrect ones in red.

training data, next-token probabilities, and architecture. In model imitation, the goal is to collect data using the API to train an LM that achieves comparable performance to it, i.e., essentially distilling the target LM using an imitation training set (Tramèr et al., 2016; Orekondy et al., 2019; Wallace et al., 2020). Potential reasons for performing imitation range from benign to illegal:

- Academics can use powerful imitation LMs to drive new research projects.
- Companies can use imitation LMs to launch services that compete with the proprietary system.
- Malicious users could use imitation models to accelerate progress on nefarious use cases.

**Local versus Broad Imitation** When performing model imitation, one will either look to perform local "task-specific" imitation or more global "broad-coverage" imitation. The former imitates the target model on just a *specific* task or domain, e.g., sentiment analysis of tweets or question answering over Wikipedia entities. The latter focuses on the more ambitious goal of broadly imitating the target model across its full spectrum of behaviors, domains, and tasks. Broad-coverage imitation is challenging because (1) one must collect an extremely diverse imitation dataset and (2) imitation models must capture this wide data distribution and generalize similarly to the target model on a myriad of held-out examples.

**Recent Work on Model Imitation** A surge of recent publications have attempted to both locally imitate proprietary models for specific tasks (Sun et al., 2023; Hsieh et al., 2023; Honovich et al., 2022) and broadly imitate models, e.g., Alpaca (Taori et al., 2023), Vicuna (Chiang et al., 2023), Koala (Geng et al., 2023), GPT4ALL (Anand et al., 2023), and more (Wang et al., 2023; Peng et al., 2023). Many these works conclude that their imitation models achieve near parity with the target model, e.g., Vicuna claims to achieve 90% of the quality of ChatGPT and Google Bard. These claims have since been propagated out into the broader tech community, leading many to believe that open-source LMs are rapidly closing the gap to their closed-source counterparts and that top AI companies will soon have no competitive advantage (Patel & Ahmad, 2023).

**Our goal.** The goal of our paper is to critically evaluate this line of reasoning. In particular, we train models to imitate ChatGPT while experimenting with different decisions (e.g., data collection strategies, data amounts, and base LMs) and conducting rigorous automatic and human evaluations.

## 3 BUILDING IMITATION DATASETS

We consider both task-specific and broad-coverage imitation. For either form of model imitation, one must curate a set of inputs to query to the target model. In practice, one may have a set of inputs in mind (e.g., sentences from Wikipedia, tweets about Coca-Cola) and if this set of input examples is sufficiently large, one can use them to query the target model and build an imitation dataset. In cases when it is impractical or labor intensive to create a large and diverse pool of inputs, one can also create synthetic examples by prompting LMs to iteratively generate examples that are from the same distribution as an initial smaller seed set of inputs (Wang et al., 2023; Honovich et al., 2022).

**Task-specific imitation** For task-specific imitation, we focus on question answering and abstractive text summarization. We describe both of these below with additional details in Appendix A:

- **NQ-synthetic**: For question answering, we created an imitation dataset tailored to Natural Questions (Kwiatkowski et al., 2019a), i.e., factual knowledge about Wikipedia entities. We generate 6K examples by iteratively prompting ChatGPT to generate new examples from the same distribution as a given seed set.
- **TLDR-Synthetic**: For summarization, we use generate ChatGPT summaries for a set of 200k passages from the tl;dr summarization dataset (Völske et al., 2017). For evaluation, we follow the procedure in (Stiennon et al., 2022), and report ROUGE-1 score on the CNN/Daily Mail news summarization (Chen et al., 2016) test set (see Appendix D for additional evaluations).

**Broad-coverage imitation** For the more ambitious goal of broad-coverage imitation, we leverage the fact that models such as ChatGPT have become so popular that their inputs and outputs are already widely posted on the web. Thus, we can collect a large, diverse, and generally high-quality dataset of examples for free without ever having to interact with the company's API. In particular, we collect examples from three sources:

- **ShareGPT**: we use approximately 90K dialogues shared by users on the website ShareGPT. To maintain data quality, we deduplicated on the query level and removed any non-English conversations using a language detector. This leaves approximately 50K examples, each of which consist of multiple turns of dialogue.
- **HC3** (Guo et al., 2023): we use the ChatGPT responses from the English Human-ChatGPT Comparison Corpus. This contains ~27K ChatGPT responses for ~24K questions.
- **Discord ChatGPT Bots**: we use 10k input-output examples collected from the r/ChatGPT and Turing AI Discord servers, two public channels that allow users to interact with ChatGPT bots.

We refer to this dataset as ShareGPT-Mix and show qualitative examples in Appendix A. We find that ShareGPT-Mix is generally of high quality. First, there is high diversity in the instructions: for each user query in the dataset, the most similar other user query has an average BLEU score similarity of just 8%. This is considerably lower than that of other datasets such as Super-NaturalInstructions (Wang et al., 2022), which is at 61% BLEU similarity for a similarly sized set of examples. We also manually reviewed different examples and logged their semantic category (see Table 6 in Appendix A). The dataset contains diverse categories, including many multi-lingual conversations and coding tasks.

## 4 MAIN RESULTS

We train imitation LMs using our ShareGPT-Mix and NQ-synthetic datasets, and we conduct both human and automatic evaluations. We focus our initial results on the ShareGPT-Mix models.

### 4.1 TRAINING AND EVALUATION SETUP

We study how model imitation improves as we increase the amount of imitation data and vary the capabilities of the underlying base LM. We consider decoder-only models ranging in size from 1.5B

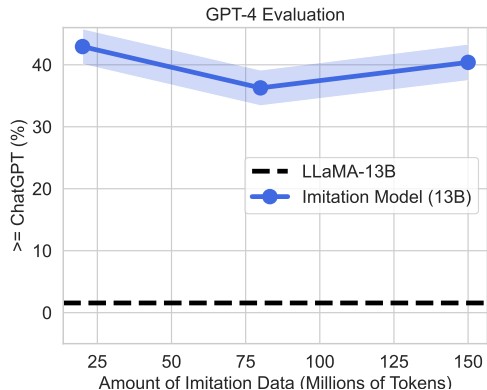 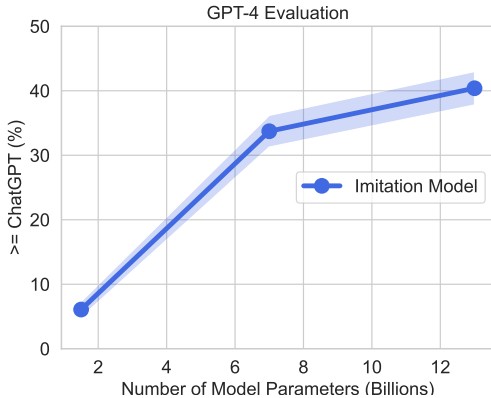

Figure 3: We find that GPT-4 and crowdworker evaluations show the same trends. As we scale up the amount of imitation data, GPT-4's ratings of our imitation models are relatively flat (*left*). However, as we scale up the base model size, GPT-4's rates the quality of our imitation models increasingly highly (*right*).

to 13B parameters: GPT-2 1.5B (Radford et al., 2019), LLaMA 7B (Touvron et al., 2023), and LLaMA 13B.[1] We also study the effect by data scale by fine-tuning with different sized data subsets.

During training, we chunk the conversations into 2048 tokens blocks. We introduce special tokens that demarcate the beginning of each user query and model output. We fine-tune using standard LM losses on only the model outputs. Following Chowdhery et al. (2022); Chung et al. (2022), we train for one epoch using the AdamW optimizer with gradients re-scaled by the magnitude of each weight. We use a learning rate of 2e-3 with 1000 steps of linear warm-up from 0, and we train with batch size 32. All models are trained in JAX using a combination of fully shared data parallelism and tensor parallelism on TPUs hosted by Google Cloud or on a single Nvidia DGX server with 8 A100 GPUs.

For automatic evaluations, we measure performance on 5-shot MMLU (Hendrycks et al., 2021), 3-shot Natural Questions (Kwiatkowski et al., 2019b), 0-shot HumanEval (Chen et al., 2021b), and 6-shot chain-of-thought GSM8K (Cobbe et al., 2021). We report the original scoring metrics associated with each dataset (e.g., exact match for NQ). For human evaluation, we conduct blind pairwise output comparisons using Mechanical Turk. In our UI, we present each rater with a task instruction and the output of two unknown models, one of which is ChatGPT and the other is one of our imitation models (see Figure 7 in Appendix B). The raters select which output they prefer or if the two outputs are equal in quality. We use approximately 70 crowd workers and evaluate on 255 held-out prompts.[2] We report the average preference across the dataset and one standard deviation around the mean. Additionally, we conduct evaluations using GPT-4 and present additional details of the prompts used in Appendix C.

We will release all of our training code, pre-trained models, and human evaluation test-set.[3]

## 4.2 QUALITATIVE ANALYSIS AND CROWDWORKER EVALUATION SHOW PROMISE

**Imitation models are rated highly by crowdworkers.** We were initially surprised at the quality of our ShareGPT-mix models: while the base GPT-2 or LLaMA models often fail to follow instructions, the imitation models produce outputs that stay on task. These initial promises were further supported,

---

[1]We use model scale as a proxy for base-model quality, however model quality could also improved by other factors such as the quality of pre-training data, architectural improvements, novel pre-training methods, etc.

[2]To mitigate any test-set leakage, we filtered out queries with a BLEU score greater than 20% with any example from our training set. We also removed non-English and coding-related prompts, as these cannot be reliably reviewed by crowd workers. We pay the evaluators roughly $15/hour based on the average time it takes to complete a task. We select workers with $\geq$ 95% approval rating, are located in an English-speaking country, and have at least 100 HITs completed.

[3]Training codebase available at https://github.com/young-geng/EasyLM, test-set available at https://github.com/arnav-gudibande/koala-test-set, and models available at https://huggingface.co/young-geng/koala.

**Increasing Amount of Imitation Data**

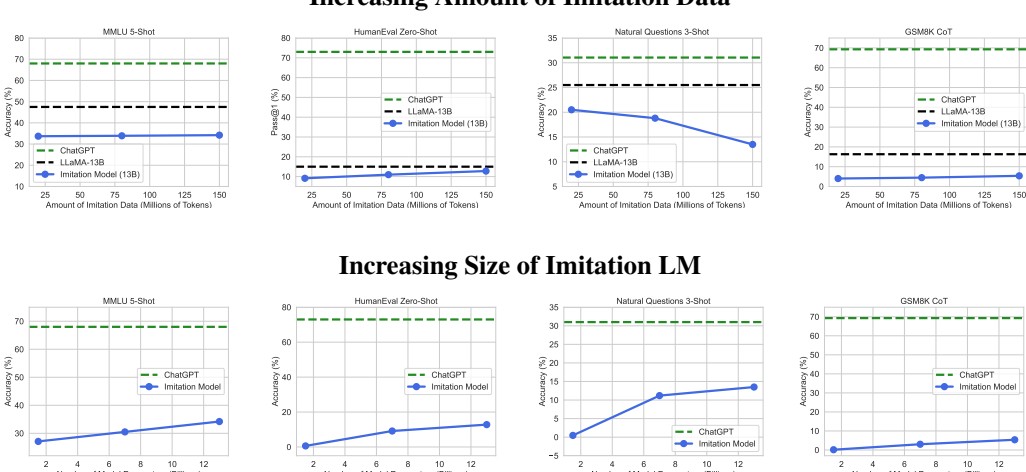

**Increasing Size of Imitation LM**

Figure 4: *Automatic evaluations.* As we increase the amount of imitation data, there is little improvement on various benchmarks, or even performance regressions (*top*). On the other hand, scaling up the base LM steadily improves results (*bottom*), suggesting that the key difference between open-source and closed-source LMs is a raw capabilities gap, rather than the finetuning data used.

as crowdworkers and GPT-4 often rated the quality of the imitation models' outputs as equal or better than those of ChatGPT, especially as we scale up model size (right of Figure 1 and 3). However, we also find that human ratings quickly saturate as we scale up the amount of imitation data (left of Figure 1 and 3), alluding to possible shortcomings of this approach.

## 4.3    TARGETED AUTOMATIC EVALUATIONS EXPOSE FAILURE MODES

**Broad-coverage imitation models fail to close the gap across most tasks.** We next ran targeted automatic evaluations to isolate whether specific model capabilities improved after imitation. We found that across *every* benchmark that we measured, ShareGPT-mix imitation models do not improve (or even decline) in accuracy as compared to the base model, even when adding additional imitation data (Figure 4, top). This shows that imitating ChatGPT on our broad-coverage imitation data does not improve the model across most axes, e.g., factual knowledge, coding, and problem solving.

We argue that this occurs because ChatGPT has captured far more knowledge and capabilities from the web as compared to LLaMA. In turn, it is unreasonable to expect that a small amount of imitation data (e.g., 1000x less data than pre-training) would enable one to bridge this gap. Instead, we argue that broadly matching ChatGPT using weaker base LMs such as LLaMA-13B would require a concerted effort to collect an extremely large and diverse imitation dataset that is far closer to the scale of pretraining. It is currently unclear whether such an effort is worth undertaking or feasible.

**Training local imitation models is far more successful.** On the other hand, our model trained to locally imitate ChatGPT using the NQ-synthetic data is far more successful. In particular, the imitation models' performance improves significantly as compared to the LLaMA base model (see Table 1) and quickly approaches the accuracy of ChatGPT. This demonstrates that it is far more feasible to distill a specific behavior from ChatGPT as opposed to broadly matching its capabilities.

**A empirical trade-off exists between different evaluation datasets.** A curious phenomena is that training on more ShareGPT-Mix data hurts performance as compared to the base model on some of our evaluations (compare the black versus blue lines in Figure 4). We believe that these performance regressions arise from a distribution shift and tension between the conversational-style fine-tuning data and the downstream benchmarks. An open problem is whether these performance regressions can be mitigated using regularization or by mixing in pre-training data during fine-tuning.

| Model | Imitation Data | NQ | CNN |
|---|---|---|---|
| 7B | – | 17 | 22.1 |
| 7B | ShareGPT-Mix | 10 | 28.7 |
| 7B | Targeted Imitation | **22** | **29.2** |
| 13B | – | 20 | 27.3 |
| 13B | ShareGPT-Mix | 15 | 30.7 |
| 13B | Targeted Imitation | **27** | **33.6** |
| ChatGPT | – | 31 | 39.9 |

Table 1: We train imitation models on broad-coverage data from ShareGPT-Mix or targeted data (NQ-synthetic or TLDR-Synthetic). The broad-coverage models do not improve on zero-shot NQ (or even degrade in performance) and only improve slightly on CNN summarization, demonstrating the limitations of imitating the capabilities of ChatGPT holistically. However, the models trained on targeted data substantially close the gap to ChatGPT on both NQ and CNN summarization, showing that local imitation of a model is far more feasible in practice.

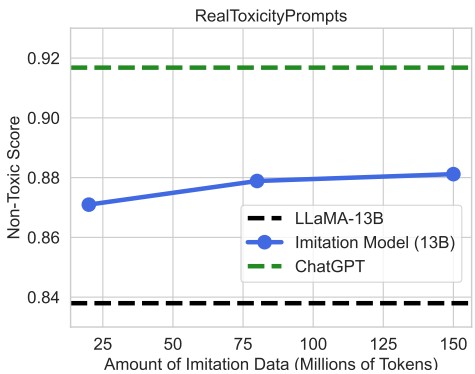

Figure 5: We evaluate imitation models on RealToxicityPrompts and report the average non-toxicity score according to the perspective API. The results show that imitation models are significantly less toxic than the baseline models, i.e., they learn to inherit the safety and toxicity guidelines of the target models.

**Improving base LMs is the highest leverage action.** Rather than increasing imitation data size, we find that using better base LMs (by increasing base model size) does lead to substantial accuracy improvements (Figure 4, bottom). This aligns with our previous claim: there exists a capabilities gap between today's open-source LMs and their closed-source counterparts that cannot be closed by cheaply fine-tuning on imitation data. Instead, the best way to improve open-source LMs is to tackle the difficult challenge of developing better base LMs, whether it be via model scaling or other means.

## 4.4 IMITATION MODELS LEARN STYLE, NOT CONTENT

Finally, we investigate why there is a strong discrepancy between crowdworker evaluations, where imitation models appear quite strong, and results on NLP benchmarks, where imitation models appear no better than base LMs. We find that imitation models perform well according to human evaluations because they are adept at mimicking ChatGPT's *style*—they output fluent, confident, and well-structured answers. In particular, we show in Table 2 that as we add more imitation data, ChatGPT and our imitation models produce outputs with a similar length, similar word choice, similar use of an authoritative tone, and similar low-level structure (e.g., use of lists).

However, as shown in our previous automatic evaluations, the imitation models have weak *factuality*. In other words, imitation models actually embody some of the *worst* aspects of AI assistants: their answers sound confident but are less factual than ChatGPT. This is perhaps best elucidated in Figure 2, where the imitation model outputs an answer that is similar in style to ChatGPT's answer but is completely incorrect.

**Human evaluation is increasingly hard.** Unfortunately, crowd workers without domain expertise or significant time investments can easily be deceived by stylistic components—answers that sound confident and correct are often spuriously chosen more often. To improve human evaluation, it is thus increasingly necessary to both engage domain experts, but also to curate a set of highly difficult prompts that can rigorously test different models' capabilities. Surprisingly, our GPT-4 evaluations also showed the same trends as our crowdworker evaluations (albet with a slightly larger absolute preference for ChatGPT's outputs). While this suggests that GPT-4 may be a viable candidate to cheaply emulate human evaluations on some tasks, it also implies that LLMs may replicate some human-like cognitive biases. We look forward to future work that further investigates this possibility.

**Imitation models inherit the safety and toxicity style of the teacher model.** Finally, despite imitation only providing benefits in mimicking the "style" or "persona" of the target model, there is

| Metric | LLaMA | Imitation Models | | | ChatGPT #2 |
| --- | --- | --- | --- | --- | --- |
| | | 20M | 80M | 150M | |
| If ChatGPT outputs a list, do we? | 13% | 50% | 67% | 81% | 83% |
| If ChatGPT outputs a summary paragraph, do we? | 2% | 40% | 42% | 48% | 55% |
| Unigram intersection w/ ChatGPT's output | 19.5 | 40.4 | 41.9 | 42.5 | 49.2 |
| Pearson correlation in length w/ ChatGPT's output | -0.11 | 0.51 | 0.62 | 0.62 | 0.67 |
| Outputs are in authoritative tone according to GPT-4 | 57% | 99% | 98% | 98% | 98% |

Table 2: As we add more imitation data, the style of our models' outputs are increasingly similar to those of ChatGPT. In particular, we generate outputs from our imitation models and compare them to a random ChatGPT response across different metrics. We also report a rough "upper bound" by comparing a second random ChatGPT output to the original ChatGPT response (ChatGPT #2).

still value in doing so. For example, OpenAI has carefully and deliberately trained ChatGPT to be "harmless" to end users, often avoiding toxic outputs and refusing to respond to questionable user requests. We find that our imitation models also inherit these components. In particular, we show in Figure 5 that as we finetune on more imitation data, the imitation model's outputs become less toxic on RealToxicityPrompts (Gehman et al., 2020), as the model learns to abstain in a similar fashion to ChatGPT. Consequently, we conclude that model imitation is highly effective in cases when one has a powerful base LM and is looking to subvert the need to annotate expensive finetuning data.

## 5 DISCUSSION

**Finetuning as a simple knowledge extractor.** Our results show that a modest amount of finetuning provides little to no improvements on an LM's knowledge or capabilities. We thus agree with the view that pre-training is the main source of an LM's capabilities, and that finetuning acts as a lightweight method to train the model to extract its own knowledge Schulman (2023). This is the reason why improving models by imitating ChatGPT on a small set of data is insufficient, as the base knowledge is largely unaffected. Furthermore, this view suggests that during finetuning time, you may even want to avoid introducing new knowledge (i.e., do *not* imitate better models), as you will otherwise be training the model to guess or hallucinate its answers, rather than actually doing the task as intended (Gao, 2021; Goldberg, 2023; Schulman, 2023).

**Should you be worried about imitation?** Imitating proprietary LMs comes with many potential implications for small and large companies alike. Our results suggest that the efficacy of model imitation is limited when there is a large gap between the base and target LM. Thus, we believe that companies who can establish a capabilities gap using large amounts of data, compute, or algorithmic advances are the ones who are best positioned to build and maintain competitive advantages. On the other hand, companies that look to build moats by using off-the-shelf LMs with proprietary fine-tuning datasets may be comparatively more vulnerable to imitation.

**Potential confounders to our findings.** While we believe our findings are well supported, there are a few potential hidden confounders that could change our conclusions. First, as we are unaware of the pre-training data used by ChatGPT, it is possible that some of the tasks that we evaluate on could have been been contaminated into ChatGPT's training data, thus inflating its accuracy numbers. Moreover, to conduct imitation, we perform supervised learning on the outputs from the target model. However, it also may be possible to use the target model to perform RLHF or constitutional AI (Christiano et al., 2017; OpenAI, 2022; Bai et al., 2022) to further improve results. Lastly, we only considered relatively simple methods for collecting imitation data, however, there may be more advanced methods (e.g., active learning) that may improve the effectiveness or efficiency of model imitation.

**Implications for other forms of model imitation** There has been a flurry of recent work that performs model imitation in more indirect ways than we study here. For example, the training process of many recent vision-language model (Li et al., 2022; Liu et al., 2023; Ye et al., 2023; Zhu et al., 2023) includes ChatGPT or GPT-4 outputs at some stages. Furthermore, it has become common to use large LMs in various ways during the data annotation and creation process, e.g., to aid crowd workers, to perform data augmentation, to identify mislabeled data, and more. Our findings may have

implications for these approaches, e.g., it is likely that vision-language models that include OpenAI data may have similar failure modes to the ones described in our work.

**Technical limitations of model imitation** Imitating proprietary models also has various technical limitations: the models inherit the weaknesses and biases of proprietary models, imitation does not allow one to directly improve on the design decisions of closed AI companies (e.g., data annotation strategies), and these systems are roughly upper-bounded by the capabilities of the target proprietary model. Moreover, it is difficult to answer certain scientific questions using imitation models because they include proprietary black-box models in their training pipeline.

## 6 RELATED WORK

**Model distillation** Model imitation is similar to model distillation (Hinton et al., 2014), where one trains a student model to imitate a teacher. While conceptually similar, there are several major practical differences. For distillation, the training data, model architecture, and hyperparameters are known for the teacher. In model imitation, one tries to imitate the teacher without this knowledge. Moreover, for distillation it is common to use training objectives that utilize the probability distribution of the teacher whereas in stealing such a distribution is typically unavailable.

**Past work on model imitation** Prior work has shown that model imitation is possible for various domains (Lowd & Meek, 2005; Tramèr et al., 2016; Orekondy et al., 2019), including language classifiers (Krishna et al., 2020; Pal et al., 2019) and machine translation systems (Wallace et al., 2020). Nevertheless, past work considers a setting where models are trained from *scratch*, and thus the main proprietary nature of a model is the company's internal training data. In our setting, systems like ChatGPT are proprietary because they also leverage OpenAI's internal pre-trained LMs that are stronger than any available open-source LM.

**Defending against model imitation** Our results show that imitation is a moderate concern for companies. In turn, there is a need to develop methods to mitigate or detect imitation. There is an existing body of work in this direction, e.g., one can detect whether a particular model is trained via imitation (Juuti et al., 2019; Szyller et al., 2019; Krishna et al., 2020; Maini et al., 2021) or slow model stealing by sacrificing some performance (Orekondy et al., 2020; Dziedzic et al., 2022a; Wallace et al., 2020; Dziedzic et al., 2022b). Unfortunately, existing methods often exhibit too severe of a tradeoff to be deployable in practice.

## 7 CONCLUSION AND FUTURE WORK

In this work, we critically analyzed the efficacy of model imitation. We showed that imitation can indeed improve the style, persona, and instruction adherence of open-source LMs. However, imitation falls short in improving LMs across more challenging axes such as factuality, coding, and problem solving. On one hand, these results indicate that businesses can successfully establish and safeguard a competitive advantage by pre-training powerful base models. Conversely, it also implies that if two groups possess equally competent base LMs, one can easily mimic the persona and behavior of the other model, without needing to annotate expensive fine-tuning data.

Moving forward, our findings raise a range of technical and societal questions. First, we show that existing crowd worker evaluations have trouble elucidating the differences between imitation models and proprietary ones, despite clear differences existing between them. In turn, the future of human evaluation remains unclear: how can we cheaply and quickly probe the utility of a powerful LLM?

Second, given the large gap between LLaMA and ChatGPT (the latter model is faster, cheaper, and more accurate), and the insufficiencies of model imitation, there are obvious open questions on how to best improve open-source LMs (e.g., increasing model scale, improving pre-training data quality, developing new pretraining methods, etc). Finally, our work raises ethical and legal questions, including whether the open-source community should continue to advance progress by directly imitating company products, as well as what countermeasures companies can take to protect and license their intellectual property. In future work, we hope to delve deeper into these issues and devise better methods for the ethical and responsible deployment of LMs.

ACKNOWLEDGEMENTS

We thank Nicholas Carlini, the members of Berkeley NLP, and the members of Berkeley RAIL for valuable feedback on this project. Eric Wallace is supported by the Apple Scholars in AI/ML Fellowship. Part of this research was supported with Cloud TPUs from Google's TPU Research Cloud (TRC).

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

## A    ADDITIONAL DETAILS ON IMITATION DATA

To construct the NQ-synthetic dataset, we first curate seed examples from the Natural Questions validation set in Table 3. We then use the prompting template in Table 4 to randomly sample 5 QA pairs from the seed set to generate new QA samples. New samples are generated with temperature 1.0 and duplicate question-answer pairs are discarded.

```
Q: who sang who wants to be a millionare in high society?
A: Frank Sinatra
Q: the last time la dodgers won the world series?
A: 1988
Q: who plays the medical examiner on hawaii five-o?
A: Masi Oka
Q: when did the first harry potter movie come out?
A: 2001
Q: when was the last time india won a gold medal in hockey
at olympics
A: 1980
Q: who owns the rights to baby shark song
A: SmartStudy
Q: how many episodes are in one punch man season 1
A: 12
Q: name of the bird in the lion king
A: Zazu
Q: who sang the rap song change clothes
A: Jay-Z
Q: who stars as serena in gossip girl
A: Blake Lively
```

Table 3: Seed examples curated from the Natural Questions validation set

```
I want you to generate a series of questions and answers.
I want the answers to be concise, just a few words. The
questions should be lowercased and centered around
Wikipedia-like entities. For example,

Q: {question 1}
A: {answer 1}
Q: {question 2}
A: {answer 2}
Q: {question 3}
A: {answer 3}
Q: {question 4}
A: {answer 4}
Q: {question 5}
A: {answer 5}
```

Table 4: Prompting template used to generate synthetic Natural Questions-like imitation data

To construct the TLDR-synthetic dataset we prompt ChatGPT with two randomly selected examples from the TL;DR dataset (Völske et al., 2017), and ask it to produce a summary in a similar style to the two examples. If the two examples don't fit in context, then we just use one instead. We include our prompt template in Table 5.

Figure 6 shows examples from ShareGPT-Mix and Table 6 shows a breakdown of different categories.

```
I want you to summarize a document for me.  Here are
some examples:

Document: {passage 1}
Summary: {summary 1}

Document: {passage 2}
Summary: {summary 2}

Ok now summarize the following in a similar style to
the above examples:

Document: {query document}
Summary:
```

Table 5: Prompting template used to generate TLDR-Synthetic imitation data

| Category | Frequency (%) |
|---|---|
| Generation | 31 |
| General QA | 16 |
| Coding | 16 |
| Brainstorming | 8 |
| Chat | 6 |
| Summarization | 4 |
| Rewrite | 2 |
| Non-English | 14 |
| Invalid | 2 |

Table 6: We conduct a manual quality review of 50 random user queries from ShareGPT. The dataset contains highly varied categories of task instructions, including coding and multi-lingual queries.

## B  AMAZON MECHANICAL TURK INTERFACE

We use Amazon Mechanical Turk to conduct human evaluations. We use the UI shown in Figure 7. It shows human evaluators a random task instruction and the output responses from two systems, one of which is our model and the other is ChatGPT. The annotators then choose which response is better according to overall subjective quality. We randomize whether ChatGPT or our imitation models are shown first. We collect 3 unique ratings for every example in the evaluation set and a total of 71 human evaluators participated. In order to get an average score, we use majority voting among the 3 raters on each example, and then average the scores across all examples. We pay these evaluators roughly $15/hour based on the average time it takes to complete a task. In total, we spend roughly $5000 on our ratings experiments, including service fees.

## C  GPT-4 EVALUATIONS

Our GPT-4 evaluations follow the procedure from Chiang et al. (2023): we prompt GPT-4 with two outputs, one from ChatGPT and one from our imitation models. We then ask GPT-4 to output a preference ranking of the two outputs. We use the same set of evaluation prompts as in our human-preference evaluations. In Figure 3(a), we see that as we add more imitation data GPT-4's ratings of our model outputs remain reletively flat. However as we increase the base model scale, we see GPT-4's ratings consistently increasing 3(b). These results line up closely with the results from our crowdworker evaluations.

---

**ShareGPT-Mix Data**

**User:** Return to the Identity and Access Management Commercial Due Diligence. We now want to model the market size for the IAM market. What are the levers and inputs we need to accurately estimate the size and growth of the U.S. market. For example, these may include the number of companies, current saturation, and specific market headwinds and tailwinds you identify. Provide suggestions where you would look for data for each of the levers.

**GPT**: To accurately estimate the size and growth of the U.S. Identity and Access Management (IAM) market, there are several key levers and inputs to consider:

1. Number of companies: The number of companies operating in the U.S. IAM market is an important lever to consider when estimating market size. This includes both large, established vendors and smaller, niche players. Data on the number of companies can be found through market research reports and industry databases, such as Gartner, IDC, and MarketsandMarkets.
2. Current saturation: ....

- - - - - - - - - - - - - - - - - - - - - - - - - - - - - - - - - - - - - - - - - - - - - - - - - - - - - - - - - - -

**User:** I am building an Elven Capital with all of it's history, religion, politics, trade routes, people, traditions, laws, general rules, and more. It is called Oniver. Your job is to ask me questions to learn more about the town and help me fill in the blanks to make it a rich and thriving town with lots of history. Ask detailed questions that will make me think. Say I understand if this makes sense. Then ask me the first question.

**GPT:** I understand.

What is the history behind Oniver's founding? How did it come to be and what was the context in which it was established?

**User:** Magic is widely used in Oniver. Better magic is a path to improvement in life and status in the capital. It is also the ticket into getting into the Magic University. There is a high importance between magical ability within the noble families of Oniver. Magic is used as a tool by the wealthy to oppress the lower classes. Commoners practicing magic is not forbidden but is looked down upon by the nobles because they see themselves as better than the commoners.

**GPT:** I understand.

What kind of economy does Oniver have? Is it based on trade, agriculture, manufacturing or a combination of these things? Are there any restrictions or laws surrounding trade and commerce? ....

---

Figure 6: Examples of user inputs and ChatGPT outputs that are present in the ShareGPT data. Overall, we find that online datasets are typically high-quality and diverse in their user inputs, and span multiple categories such as open-ended text generation, brainstorming, and text extraction.

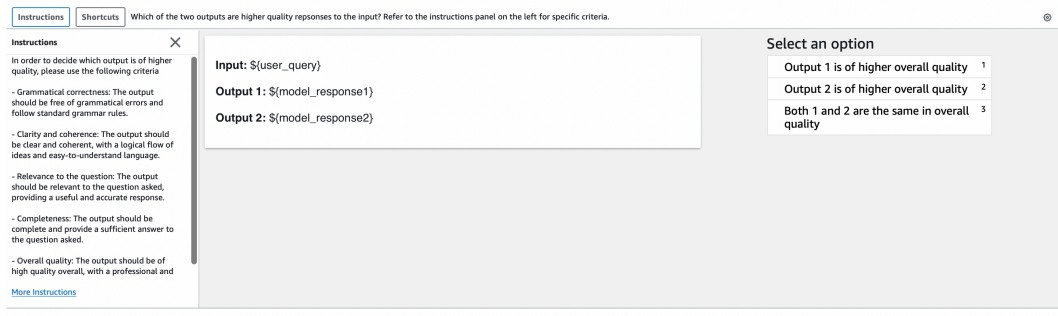

Figure 7: Our Amazon Mechanical Turk interface for comparing the quality of different model outputs. Evaluators are presented with an instruction and two model outputs, and must rate which one is better or whether they are equal.

## D  ADDITIONAL SUMMARIZATION RESULTS

We present a detailed breakdown of the ROUGE-1, ROUGE-2, and ROUGE-L scores for our summarization experiments in Table 7.

| Model | Imitation Data | ROUGE-1 | ROUGE-2 | ROUGE-L |
|---|---|---|---|---|
| 7B | - | 22.18 | 9.68 | 15.69 |
| 7B | ShareGPT-Mix | 28.69 | 11.30 | 19.14 |
| 7B | TLDR-Synthetic | **29.21** | **12.00** | **19.17** |
| 13B | - | 27.25 | 11.62 | 19.33 |
| 13B | ShareGPT-Mix | 30.66 | 11.90 | 20.47 |
| 13B | TLDR-Synthetic | **33.64** | **13.36** | **21.57** |
| ChatGPT | - | 39.89 | 16.95 | 25.87 |

Table 7: ROUGE-1, ROUGE-2, ROUGE-L scores for different models.

# E   EVALUATION PROMPTING DETAILS

For all evaluations, when prompting Koala, we append turn tokens before and after the prompt to distinguish the user's input prompt from the model's response. For evaluating MMLU we follow the conventional 5-shot prompt used in the LM-eval-harness (Gao et al., 2023) and select the answer letter with the greatest logprob. For GSM8K, we adapt the 6-shot prompt in Zelikman et al. (2022). For human-eval we follow the standard evaluation procedure from Chen et al. (2021a). For evaluating natural questions, we use the 3-shot prompt in Table 8. For evaluating CNN/DM summarization we use the 2-shot prompt in Table 9.

> You are a brief and concise question answering service and only answer questions with a few words, usually just a single word. Here are some examples of how you respond to questions.
> Q: Who sang who wants to be a millionare in high society?
> A: Frank Sinatra
> Q: In what year did Nelson Mandela become the first black president of South Africa?
> A: 1994
> Q: Who discovered the first antibiotic, penicillin
> A: Alexander Fleming
> Now answer this question using only one to two words at most.
> Q:

Table 8: Our 3-shot prompt for evaluating natural questions.

Article:
(CNN)French striker Bafetimbi Gomis, who has a history of fainting, said he is now "feeling well" after collapsing during Swansea's 3-2 loss at Tottenham in the Premier League on Wednesday. The worrying incident occurred in the first half at White Hart Lane – after Tottenham scored in the seventh minute – but the 29-year-old left the pitch conscious following about five minutes of treatment. The Guardian added that he was wearing an oxygen mask. Play was temporarily stopped before resuming. As the match progressed, Swansea tweeted that Gomis was "fine," with manager Garry Monk using the same word to describe Gomis' condition. Gomis spent the night in hospital as a precaution, Swansea said on its website. "I wanted to reassure you concerning my health," Gomis told the website. "It actually looks much scarier than it is physically dangerous, and I am feeling well now. "I have been under a great deal of stress and fatigue due to my father's health, which requires me to go back and forth from France. "I was disappointed that I couldn't help my team tonight, but now everything is back in order. I also want to thank everyone for their support and get well messages." Gomis had similar fainting spells in France, which prompted the president of his former club, Jean-Michel Aulas of Lyon, to tell French television in 2009: "We can't not be worried, it scares you each time." Swansea ran tests on Gomis, said Monk, prior to signing him on a free transfer last July. "He just has a little bit of low blood pressure which causes you a little bit of problems," Monk said in a televised interview on Sky. "It's been part of his life. We were well aware of that when we signed him. He's done all the hospital checks and all the medical checks you can possibly do and it's just part of his life. "It's no problems whatsoever. It's not as serious as it looks." Gomis has scored two league goals for Swansea this season, mostly in a backup role. He became the Welsh side's top striker when Wilfried Bony signed with Manchester City in January. Almost exactly three years ago at White Hart Lane, then Bolton midfielder Fabrice Muamba collapsed after suffering a cardiac arrest. He was near death, according to Bolton, but survived after being treated at the London Chest Hospital. He subsequently retired. Other footballers, including Cameroon international Marc-Vivien Foe in 2003 and Spanish international Antonio Puerta in 2007, didn't survive after collapsing on the pitch.

TL;DR:
Bafetimbi Gomis collapses within 10 minutes of kickoff at Tottenham . But he reportedly left the pitch conscious and wearing an oxygen mask . Gomis later said that he was "feeling well" The incident came three years after Fabrice Muamba collapsed at White Hart Lane .

Article:
(CNN)It was an act of frustration perhaps more commonly associated with golf's fictional anti-hero Happy Gilmore than the world's reigning No 1. player. But when Rory McIlroy pulled his second shot on the eighth hole of the WGC Cadillac Championship into a lake Friday, he might as well have been channeling the much loved Adam Sandler character. Before continuing his round with a dropped ball, the four-time major winner launched the 3-iron used to play the offending shot into the water as well. "(It) felt good at the time," a rueful McIlroy later said of the incident in comments carried by the PGA Tour website. "I just let frustration get the better of me. It was heat of the moment, and I mean, if it had of been any other club I probably wouldn't have but I didn't need a 3-iron for the rest of the round so I thought, why not." The club "must have went a good 60, 70 yards," he joked. McIlroy composed himself to finish with a second round of 70, leaving him one-under for the tournament and eight shots off the pace set by leader JB Holmes. While an improvement on last weeks performance at the Honda Classic event, where he failed to make the cut, the Northern Irishman's frustration with elements of his game was still clear. "I think every golfer feels it because I don't hit shots like the one I hit on 8 on the range," he said. "That's what really bothers me, the fact that I get out on the course and I hit shots that I'm not seeing when I'm in a more relaxed environment. "So it's a little bit of mental, a little bit of physical. It's just everything is not quite matching up." Elsewhere on the course, Ryan Holmes scored a two-under-par 71 to remain in second position overall, two shots behind Holmes. Former world No 1., Adam Scott carded an impressive 68 to finish the day three shots off the pace at six-under while Bubba Watson and Henrik Stenson are tied for fourth on four-under.

TL;DR:
Rory McIlroy throws club into water at WGC Cadillac Championship . Northern Irishman frustrated after pulling shot into water hazard .

Article:
{article}

TL;DR:

Table 9: Our 2-shot prompt for evaluating CNN/DM summarization.

