# OpenReview forum: "The False Promise of Imitating Proprietary Language Models"
_ICLR.cc/2024/Conference — ICLR 2024 spotlight_

### Official Review · Reviewer_WxVs · 2023-10-22

**Soundness:** 3 good
**Presentation:** 3 good
**Contribution:** 3 good
**Rating:** 8
**Confidence:** 4

**Summary:**

The authors critically investigate the promise of finetuning language models using (imitation) data obtained from more capable language models(LMs). The paper challenges the assumption that this process improves an LM overall – but suggests that this is rather `mimicking their style`. They propose several interesting findings around what aspects of the performance improve or deteriorate upon tuning LMs on imitation data.

**Strengths:**

I overall support the messages and findings in the paper.

1. I appreciate this timely and critical investigation. Finetuning existing LMs on instruction-tuning data from more capable LMs has swiftly become common practice, yet we do not fully understand the change in behavior. The paper raises several critical questions that I would personally appreciate having in the literature.

2. One novel and important finding is that the authors find that even though training on imitation data improves the results in crowd worker evaluations, they observe even a degradation in factuality. This is a significant consideration that should be kept in mind when finetuning models. This further raises a question about what other aspects of capabilities may be fluctuating during imitation data training.

3. On the other hand, they inherit some of the useful properties, such as reduced toxicity / being more safe. Again, this further informs us about what really happens when models are trained on imitation data. It would be compelling to further explore a more fine-grained decomposition of performance gains and losses upon training on imitation data.

4. The experiments are large-scale and informative yet not cheap to perform, thus findings enable valuable conclusions that are otherwise not easy to draw.

**Weaknesses:**

There are 2 main points that I am concerned about.

1. (IRB Approval / Exemption) The human study in the paper does not seem to have the relevant IRB approval or an exemption. The code of ethics states `Where human subjects are involved in the research process (e.g., in direct experiments, or as annotators), the need for ethical approvals from an appropriate ethical review board should be assessed and reported.` I am deferring the judgment about this to Ethics Reviewers / ACs. This should have most probably been done before the human subject study, but I encourage the authors to swiftly go through the IRB process for clarity.

2. (Definition of Capability) The coverage of the capability evaluations is somewhat limited. Currently, authors evaluate mostly on factuality tests like MMLU/NQ/HumanEval and also show results around toxicity but how about other capabilities? Could it be that the imitation data leads LMs to reason better? Or, could it be that imitation data drives better calibration? While I do understand that there is a finite compute budget and there should be a limit in evaluation, the capability definition here is rather to limit the conclusions to draw.

3. (Minor, Title) Given my concern in 2, I’m personally slightly skeptical to call this a `false promise`. It is unclear if it is broadly a false promise, or if there are capabilities that are improved – it’s rather there exists capabilities that this process even hurts, and we should be mindful about trusting crowdsourcing or other automated evaluations to understand the impact of a process.

**Questions:**

1. The discussion seems to rely heavily on the concept of `tuning of imitation data`. However, I think one distinction is the kind of imitation data used to finetune most of the models, which are usually based on roughly arbitrary conversations between users and models. Would the authors agree that if the imitation data is constructed in a different way, then it may be possible to improve e.g. factuality of the finetuned model? For instance, I can imagine the way to construct imitation data to be around extracting rare knowledge, and then possibly the finetuned model could be improved a lot.

2. Have the authors explored other capability definitions than factuality and toxicity? For instance, do we know anything about reasoning, calibration, creativity, truthfulness.. ?

3. Did the authors get the necessary approvals from the IRB of their institution?

**Details Of Ethics Concerns:**

The study involves human subjects (through MTurk), and it is unclear to me whether the authors got the necessary IRB approvals.

---

> ### Author Response · Authors · 2023-11-20
> **Author Response to Rebuttal**
>
> We thank the reviewer for their helpful comments and suggestions and are glad that they believe our findings to be novel, important, and timely!
>
> We respond to the main concerns below:
>
> ```
> (IRB Approval / Exemption) The human study in the paper does not seem to have the relevant IRB approval or an exemption. The code of ethics states Where human subjects are involved in the research process (e.g., in direct experiments, or as annotators), the need for ethical approvals from an appropriate ethical review board should be assessed and reported. I am deferring the judgment about this to Ethics Reviewers / ACs. This should have most probably been done before the human subject study, but I encourage the authors to swiftly go through the IRB process for clarity.
> ```
>
> We have an irb for “Chatbots for Task-Oriented Dialogue” Protocol # 2021-04-14272. We will include this in our acknowledgements section of the camera ready paper.
>
> ```
> (Definition of Capability) The coverage of the capability evaluations is somewhat limited. Currently, authors evaluate mostly on factuality tests like MMLU/NQ/HumanEval and also show results around toxicity but how about other capabilities?
> ```
> ```
> Have the authors explored other capability definitions than factuality and toxicity? For instance, do we know anything about reasoning, calibration, creativity, truthfulness.. ?
> ```
>
> We agree that there are many additional questions that one could potentially ask about model imitation, and many of these could be a great subject for future work. In our paper we focused on a representative set of LM evaluations targeting factuality, coding ability, toxicity, and human preference. However, we agree that one notable exception from this list is reasoning. For that reason, we will add evaluations on gsm8k to our camera ready paper.
>
> ```
> (Minor, Title) Given my concern in 2, I’m personally slightly skeptical to call this a false promise. It is unclear if it is broadly a false promise, or if there are capabilities that are improved
> ```
>
> We agree with the sentiment of this comment and plan to adjust the framing of the title and abstract for the camera ready version of the paper.
>
> ```
> Would the authors agree that if the imitation data is constructed in a different way, then it may be possible to improve e.g. factuality of the finetuned model?
> ```
>
> Yes, we would agree with this statement. In fact, there are some more recent works [1,2] which use novel methods for generating imitation data, and they show some signs that improving factuality may be possible.
>
> [1]: Orca: Progressive Learning from Complex Explanation Traces of GPT-4
>
> [2]: WizardLM: Empowering Large Language Models to Follow Complex Instructions

---

> > ### Comment · Reviewer_WxVs · 2023-11-21
> > **Response to authors**
> >
> > Thank you for the rebuttal!
> >
> > > IRB
> >
> > Does the protocol number indicate it was approved in 2021, or is that just an arbitrary number? I'm not sure what is the situation in your institution, however, there are many institutions where IRB approvals are valid for only 1 year. Do you suggest that this is not the case with your institution and that the details of your human study protocol were included in 2021?
> >
> > I acknowledge the author's responses, and I do not have other questions. I do support this paper and its messages, however, we should be clear about what the suggested IRB entails and the authors practice a responsible study.

---

> > > ### Author Response · Authors · 2023-11-22
> > >
> > > Thanks for checking in. I realized a mistake in the previous response, the protocol id is 2022-07-15514. It has not yet expired. The IRB is for creating "chatbots for better dialogue": "This study investigates the use of machine learning to build a chatbot that can get better at responding to a human through several interactions."
> > >
> > > However, even though we have an IRB, we would like to note that in this case our crowd-worker ratings are not studying human subjects, they are studying AI models, and thus an IRB is not technically a strict requirement for this work.

---

### Official Review · Reviewer_A4hH · 2023-10-26

**Soundness:** 2 fair
**Presentation:** 2 fair
**Contribution:** 4 excellent
**Rating:** 6
**Confidence:** 4

**Summary:**

The paper critically analyzes the approach of imitating proprietary systems (e.g., ChatGPT) by finetuning LMs using various model sizes, data sources, and imitation dataset sizes. Then the authors do human evaluation as well as evaluation on NLP benchmarks. Imitation models do not do well on tasks not heavily supported by the imitation data.

One interesting finding is that training on broad-coverage imitation data may decrease Natural Questions factuality, but training on NQ-like-data only will increase the accuracy.

The authors also conclude that the best action forward is to improve base LMs, instead of doing imitation on proprietary systems.

**Strengths:**

I vaguely heard of this paper when it came out but it’s my first time reading it. The motivation is excellent for sure (the public will care about this paper), given that many groups and startups are imitating proprietary language models potentially as a shortcut.

The findings are useful to many practitioners -- they'll likely carefully think whether knowledge distillation is useful or how it'll be useful.

Some findings are quite interesting (see summary above for example).

**Weaknesses:**

I have three concerns related to crowdsourcing (see the next three paragraphs).

Do human raters have low quality? The incentive design and crowdworker filtering seem lacking.
- What’s the human agreement (e.g., Fleiss' kappa)?
- What’s the average time humans spend on each comparison?
- Is there an option for humans to decline the comparison (because they may not be knowledgeable enough)?
- How do you make sure that humans are rewarded based on correct choices, and potentially punished if they do extremely poorly?

It’d be useful to have human evaluators write out rationales on why they chose one over another (or rate on multiple scales using multiple metrics). Otherwise concluding “human evaluators rate imitation models’ outputs higher because of their style” seems only a conjecture to me.

If crowdworkers have low quality (thus their annotations unreliable), then it doesn't seem prudent to use Figure 1(c) (crowdworker preference vs. number of model parameters) to reach the conclusion that we should improve base LLMs.


I also have some other concerns:

There are two settings for imitation in the paper. The second setting is broad-coverage imitation. The imitation dataset size could be much larger. Currently the authors are using around (90+27+10)K examples, but this is quite a small number of examples – the dataset size is even smaller than most of the machine translation training sets from ten years ago.


The results in this paper are only specific to supervised fine-tuning, not RLHF for example. This should be qualified in the intro paragraph.

The authors claim that matching ChatGPT using imitation would require an “enormous” amount of imitation examples. Is this supported anywhere in the paper?



Update: I read the authors' response. I'm still a bit concerned about human annotation quality, but I raised the score.

**Questions:**

Important: What are the decoding parameters for each model? (This is not addressed post-rebuttal.)


Below are minor (or very minor issues) in evaluating this paper:

I wonder if practitioners may interleave ChatGPT imitation data with actual pretraining data and fine-tuning data. It’s unclear if this setting would lead to the same problems.

The base models discussed in this paper are quite weak. For example, llama-1-13b is used instead of the SFT- and RLHF-tuned llama-2-13b-chat. The models are quite small too. Unclear if the results generalize to larger models.

---

> ### Author Response · Authors · 2023-11-20
> **Author Response to Rebuttal**
>
> We thank the reviewer for their helpful comments and suggestions and are glad that they believe that our motivation is excellent and that our findings are useful and interesting!
>
> We respond to the main concerns below:
>
> ```
> Do human raters have low quality? The incentive design and crowdworker filtering seem lacking
> ```
>
>
> We will include further details on our human evaluation setup in the camera ready version of the paper. While we agree that many more sophisticated and costly human rater setups are possible, we and many others in the community use this setup or setups similar to ours to obtain human ratings for model outputs. Our point is not that humans in general can't distinguish between outputs from models of different capability levels but that many existing crowd worker evaluations are unable to distinguish these. This underscores our point that LM evaluation is a challenging open problem for future work to address.
>
> ```
> If crowdworkers have low quality (thus their annotations unreliable), then it doesn't seem prudent to use Figure 1(c) (crowdworker preference vs. number of model parameters) to reach the conclusion that we should improve base LLMs.
> ```
>
> We are not relying on Figure 1(c) to reach this conclusion, we are relying on an ensemble of evidence including the quantitative results on several benchmarks across several model sizes.
>
> ```
> There are two settings for imitation in the paper. The second setting is broad-coverage imitation. The imitation dataset size could be much larger. Currently the authors are using around (90+27+10)K examples
> ```
>
> We agree that training on larger datasets is indeed possible, but since we find that even increasing data by roughly an order of magnitude yields very few returns, we believe our conclusions hold.
>
>
> ```
> The authors claim that matching ChatGPT using imitation would require an “enormous” amount of imitation examples. Is this supported anywhere in the paper?
> ```
>
> We believe that this is supported by our scaling curves, but we will make this statement more precise in the camera ready version of our paper.
>
> ```
> I wonder if practitioners may interleave ChatGPT imitation data with actual pretraining data and fine-tuning data. It’s unclear if this setting would lead to the same problems.
> ```
>
> Continuing to train on many more tokens of pre-training data is indeed a way to improve the base model, which would be inline with our conclusions about improvising base pretrained models. Therefore, we believe that this is out of scope for our paper, but could be interesting research for subsequent work.

---

> > ### Author Response · Authors · 2023-11-22
> >
> > Just checking in, the discussion period ends later today. We'd appreciate any further comments or feedback you have. Thanks!

---

### Official Review · Reviewer_TcvA · 2023-11-01

**Soundness:** 3 good
**Presentation:** 4 excellent
**Contribution:** 4 excellent
**Rating:** 8
**Confidence:** 4

**Summary:**

The paper critically analyzes the method of using the output of a stronger LM to fine-tune and improve a weaker LM, pointing out that model imitation is not a free lunch.
The authors concluded that broadly matching ChatGPT using purely imitation would require (1) a concerted effort to collect enormous imitation datasets and (2) far more diverse and higher quality imitation data than is currently available.

**Strengths:**

1. Using the output of GPT-4 to cheaply improve a weaker language model by fine-tuning is widely adopted. This paper analyzes the drawbacks of doing so, which is helpful to guide the direction of developing more powerful LLMs.

2. A large number of experiments and analyses prove the author's point of view.

**Weaknesses:**

1. The author claimed that it is far more feasible to distill a specific behavior from ChatGPT as opposed to broadly matching its capabilities. However, the paper only conducted experiments on NQ-synthetic data.

2. The paper claimed that imitation models are adept at mimicking ChatGPT's style but not its factuality and become far better at following instructions. But the other important ability of the model, that is, the ability to reason, has not been well studied.

**Questions:**

Is model imitation still a good solution if the model's factual performance is decoupled to the retrieval model?

---

> ### Author Response · Authors · 2023-11-20
> **Author Response to Rebuttal**
>
> We thank the reviewer for their helpful comments and suggestions and are glad that they think our work is “helpful to guide the direction of developing more powerful LLMs”.
>
> We respond to the main concerns below:
>
> ```
> The author claimed that it is far more feasible to distill a specific behavior from ChatGPT as opposed to broadly matching its capabilities. However, the paper only conducted experiments on NQ-synthetic data.
> ```
>
> We will add an additional experiment on task-specific distillation for abstractive text summarization to the camera ready version of our paper.
>
> ```
> The paper claimed that imitation models are adept at mimicking ChatGPT's style but not its factuality and become far better at following instructions. But the other important ability of the model, that is, the ability to reason, has not been well studied.
> ```
>
> We agree that the effectiveness of model imitation for learning reasoning is indeed an interesting question! Some recent work has studied this [1]. However, we will also add evaluations on gsm8k to test this question ourselves.
>
> [1]: Orca: Progressive Learning from Complex Explanation Traces of GPT-4

---

> > ### Comment · Reviewer_TcvA · 2023-11-22
> >
> > Thank you for your reply. Could you provide some insights into the Questions ？

---

> > > ### Author Response · Authors · 2023-11-22
> > >
> > > Yes! To answer the question:
> > > ```
> > > Is model imitation still a good solution if the model's factual performance is decoupled to the retrieval model?
> > > ```
> > >
> > > We believe that this is a very interesting question to investigate! It is indeed very plausible that model imitation may behave differently when paired with retrieval. Though we believe this question is out of scope for our work. In general, there are many additional questions that one could potentially ask about model imitation, and many of these could be a great subject for future work. In our paper we focused on studying the behavior of model imitation in a setting similar to prior work [1,2,3], so as to better understand model imitation's impact on a representative set of LM evaluations targeting factuality, coding ability, toxicity, and human preference.
> > >
> > > [1]: Vicuna: An Open-Source Chatbot Impressing GPT-4 with 90%* ChatGPT Quality
> > >
> > > [2]: Koala: A Dialogue Model for Academic Research
> > >
> > > [3]: Alpaca: A Strong, Replicable Instruction-Following Model

---

### Official Review · Reviewer_msgU · 2023-11-01

**Soundness:** 2 fair
**Presentation:** 3 good
**Contribution:** 2 fair
**Rating:** 6
**Confidence:** 3

**Summary:**

The authors investigate the question of acheiving performance parity with high-quality proprietary systems by training (smaller, generally lower quality models) on the outputs of the proprietary systems. The investigation is carried out over a range of data sizes collected from proprietary systems, or imitation data, and a range of model sizes. The authors' conclude that while training on some imitation data can improve the style of weaker models, there (i) is still a large performance gap to the proprietary models especially in evaluations of general capabilities, (ii) diminishing returns from increasing imitation data, (iii) greater gains (than collecting imitation data) by simply increasing the model size.

**Strengths:**

The main strengths of this work:
- The question studied is important as most open-source models make use of imitation data for supervised finetuning.
- The investigation along the data and model size axes is well thought out.

**Weaknesses:**

The main weaknesses of this work:
- The implicit assumption of this work (revealed in the title) is that there exists a claim or understanding that imitating proprietary language models by sampling their outputs for training is all that is needed to achieve performance parity - however, I contend that this isn't the prevalent understanding. It is understood that proprietary model output is a good source of finetuning data but not necessarily the only source. See for example the use of FLAN alongside imitation datasets like Alpaca for SFT.
- The authors present style imitation of propreitary models as a negative aspect of training on imitation data (at least in the abstract), however the right amount of style imitation can be a definite source of improvement for open-source models - as the authors point out in Section 4.4. My suggestion would be to revise the abstract to reflect this more accurately.

**Questions:**

In Figure 4. the 5-shot MMLU performance of the 13B imitiation model is quite low for a model of that size - can the authors describe the setup in more detail?

Some discussion of the points raised in "weaknesses" would also be welcome.

---

> ### Author Response · Authors · 2023-11-20
> **Author Response to Rebuttal**
>
> We thank the reviewer for their helpful comments and suggestions and we are glad that they believe that our work is important and that our experimental setup is well thought out!
>
> We respond to the main concerns below:
>
> ```
> The implicit assumption of this work (revealed in the title) is that there exists a claim or understanding that imitating proprietary language models by sampling their outputs for …
> ```
>
> We agree that there exist many alternative sources of high quality finetuning data which do not happen to be generated by proprietary language models, and that these sources may imbue different properties in the source model than the datasets that we consider in our paper.
>
> Additionally, we agree that the prevailing understanding of model imitation and more broadly finetuning has quickly evolved in recent months and that this may not be the prevalent understanding today. We will therefore reframe the abstract and title to better reflect this for the camera ready.
>
> ```
> The authors present style imitation of propreitary models as a negative aspect of training on imitation data (at least in the abstract), however the right amount of style imitation can be a definite source of improvement for open-source models - as the authors point out in Section 4.4.
> ```
>
> We agree that there are many benefits to model imitation and will update our abstract and title to more clearly reflect this in the camera ready version.
>
> ```
> In Figure 4. the 5-shot MMLU performance of the 13B imitation model is quite low for a model of that size - can the authors describe the setup in more detail?
> ```
>
> We will add more details about our evaluation procedures to the appendix, but to answer your question, we use a standard MMLU evaluation procedure based on the [lm-evaluation-harness](https://github.com/EleutherAI/lm-evaluation-harness) repo with one exception. For our imitation models we abalated evaluations using both the log-probs of answer letters to select the answer choice and using the log-probs of the answer text. We found that using answer-letters consistently yielded higher MMLU performance on our imitation models, so we report the results using answer-letters in the paper.

---

> > ### Comment · Reviewer_msgU · 2023-11-22
> >
> > Thanks for the reply - great to hear you acknowledge my concerns.
> >
> > Can you elaborate on how you are thinking of reframing this work?

---

> > > ### Author Response · Authors · 2023-11-23
> > >
> > > We have adjusted our abstract to read:
> > >
> > > ```
> > > ... Overall, we conclude that while model imitation can be useful for training models to follow instructions and avoid toxic outputs, it falls short its full promise in many ways. In particular, there exists a substantial capabilities gap between open and closed LMs that we find cannot be bridged merely by adding more imitation data. Instead, we find that fine-tuning more capable base LMs has a significantly more substantial effect on closing this gap. In turn, we argue that the higher leverage action for improving open-source models is to tackle the difficult challenge of developing better base LMs, rather than taking the shortcut of imitating proprietary systems.
> > > ```
> > >
> > > We believe that this new abstract explicitly clarifies the possible benefits of model imitation and adjusts some of the "false promise" rhetoric, further alluding to the fact that model imitation is not all bad. Additionally, we believe this new abstract more clearly states our findings. Hopefully these adjustments help to abate some of your concerns. We are also brainstorming new title options, and will modify the title similar to the abstract in the camera ready.

---

> > > > ### Comment · Reviewer_msgU · 2023-11-23
> > > >
> > > > With the revised abstract and title more accurately reflecting the takeaways from this work, I am willing to increase my score to 6.
> > > >
> > > > However, I still believe we (as a field) are moving towards unlocking ways to improve model training on the outputs of larger proprietary models and it is too early to conclude the negative result. If accepted, this paper would serve as a historical note and an example of where model imitation breaks down.

---

### Public Comment · ~Zechuan_Hu1 · 2023-12-07
**The impact of data diversity**

I basically agree the findings in this work. However, i have some questions.
1. Have you explore the impact of data diversity in imitating proprietary LLMs? Increasing the amount of imitation data may not lead the imitation LLM to learn how to handle some specific tasks, e.g. NQ-questions (in other words, the imitation LLM may not learn the response style?). In figure 1, is it possible that score increase as the data diversity increase?
2. "Training local imitation models is far more successful."
     -- Is the finding suitable for other specific tasks, e.g., reasoning, math?
     -- For the success reason, is it possible that the NQ data injects external knowledge to the imitation LLM?

---

### Meta-Review · Area_Chair_xEWs · 2023-12-12

**Metareview:**

**Paper Summary:**

This paper presents a critical analysis of the practice of fine-tuning weaker language models (LMs) using outputs from stronger, proprietary systems like ChatGPT. The authors conduct extensive experiments using various model sizes and imitation data amounts, evaluating their results through crowd raters and NLP benchmarks. They find that while these imitation models initially appear competitive with ChatGPT in terms of style and instruction following, a deeper analysis reveals a significant gap in general capabilities and factuality compared to the base model. The paper concludes that imitation as a strategy is insufficient for closing the performance gap between open and closed LMs and advocates for focusing on developing better base LMs.

**Strengths:**

1. Relevance and Importance: The study addresses the use of imitation data for fine-tuning LMs. This is particularly relevant given the widespread use of such practices in open-source model development (msgU, TcvA, A4hH, WxVs).
2. Comprehensive Experiments: The investigation spans a broad range of data and model sizes, providing a thorough evaluation of the effectiveness of imitation in different contexts (msgU, TcvA, WxVs).
3. Valuable Insights: The paper uncovers important findings, such as the limited benefits of style imitation and the potential trade-offs in factuality when using imitation data (TcvA, WxVs).
4. Critical Analysis of Current Practices: The paper challenges the current trend of fine-tuning existing LMs on instruction-tuning data from more capable models, contributing to the broader discourse on the development of powerful LLMs (TcvA).

**Weaknesses:**

1. Limited Capability Evaluation: The paper's focus is primarily on factuality, with less attention to other capabilities like reasoning, calibration, and creativity. Expanding the scope of evaluation could provide a more comprehensive understanding of the impacts of imitation data (WxVs, TcvA).
2. Generalizability of Findings: The findings are specific to supervised fine-tuning and may not apply to other training approaches like RLHF. Additionally, the applicability of results to larger models is uncertain (A4hH).


**Decision:**

This paper evaluates current practices of distilling proprietary LMs, which contributes to the understanding of the limitations of distillation in model training and encourages a shift towards improving base LMs. Therefore, I recommend accepting this paper as a spotlight.

**Justification For Why Not Higher Score:**

If space permits I wouldn't be against accepting this as an oral.

**Justification For Why Not Lower Score:**

This paper addresses the important question of whether we can imitate proprietary LMs through distillation, which is a very timely contribution and I think it deserves at least a spotlight.

---

### Decision · Program_Chairs · 2024-01-16

Accept (spotlight)